# Associations of Body Condition Score at Calving, Parity, and Calving Season on the Performance of Dairy Cows and Their Offspring

**DOI:** 10.3390/ani13040596

**Published:** 2023-02-08

**Authors:** Milaine Poczynek, Larissa de Souza Nogueira, Isabela Fonseca Carrari, Jorge Henrique Carneiro, Rodrigo de Almeida

**Affiliations:** Department of Animal Science, Universidade Federal do Paraná, Rua dos Funcionários 1540, Curitiba 80035-050, Paraná, Brazil

**Keywords:** body reserves, dairy calves, fetal programming, transition period

## Abstract

**Simple Summary:**

This prospective study aimed to evaluate the impact of body condition score (BCS) at calving, parity, and the calving season on the performance of dairy cows and their offspring. Primiparous offspring were born lighter but achieved similar performance of the other parities and had less culling until the first calving. Calves born in the winter were heavier at birth, calved younger, and produced more milk at first lactation. Calves born from lower BCS cows were lighter and had more culling until weaning.

**Abstract:**

This study aimed to evaluate the impact of body condition score (BCS) at calving, parity, and the calving season on the performance of dairy cows and their offspring. Data from 521 Holstein cows that calved a female calf and had their BCS evaluated at calving from a single commercial farm located in Southern Brazil were used. Cows were categorized into five BCS classes: class 1: <3.0 (*n* = 19), class 2: 3.0–3.25 (*n* = 134), class 3: 3.5–3.75 (*n* = 160), class 4: 4.0–4.25 (*n* = 142), and class 5: >4.25 (*n* = 66). Data were also categorized by calving order (primiparous and multiparous dams) and by calving season. The study was designed as a prospective cohort study. Variables with normal distribution were analyzed by the MIXED procedure of SAS, while binary outcomes were analyzed by the GLIMMIX procedure of SAS. Daughters from primiparous dams were born lighter (39.1 ± 0.42 vs. 41.4 ± 0.29 kg, *p* < 0.01), but they had the same weights as the daughters from multiparous cows at weaning (121.5 ± 1.67 vs. 120.4 ± 1.58 kg, *p* = 0.20). As expected, primiparous cows showed lower (*p* < 0.01) 305-day milk yields than multiparous ones: 8633 ± 363 vs. 10,761 ± 249 kg, respectively. Regarding the calving season, cows that calved in the winter were the most productive ones, and those that calved in the fall had lower milk yields (*p* = 0.01). Calves born in the winter were heavier at birth (*p* < 0.01), calved younger (*p* = 0.04), and produced more milk at first lactation (*p* = 0.03). The BCS class had an impact (*p* < 0.01) on calf birth weights; daughters from Class 1 cows (BCS < 3.0) were lighter (38.0 ± 1.0 kg) than the calves from Class 5 cows with a BCS > 4.25 (41.9 ± 0.57 kg). Calves from dams with a BCS < 3 (Class 1) had a 31.8% culling rate until weaning, while calves from cows with a BCS of 3.0–3.25 (Class 2) had a 9.6% culling rate (*p* = 0.12). These results suggest that maternal and environmental factors, such as calving season and parity, in addition to the dams’ body condition score at calving, are associated with different offspring performances.

## 1. Introduction

Efficient replacement cattle rearing is important for the sustainability of dairy farms. However, high morbidity and mortality rates are commonly observed in some dairy farms. After leaving the uterine environment, calves experience many challenges in the external environment, such as exposure to a wide variety of pathogens, metabolic adaptations, and thermal regulation, when they still do not have a well-developed and functional immune system.

Suboptimal intrauterine conditions during critical periods of development lead to changes in tissue structure and function such as number and metabolism of cells and activation or not of specific genes. That may have long-term consequences on the offspring’s physiology and disease susceptibility. Indeed, pre-weaning metabolic function and growth are associated with future milk production [1].

The body condition score (BCS) measurement is an important tool for the nutritional management of dairy farms [2] and allows the estimation of adipose tissue reserves in cows [3]. The evaluation method was proposed by Wildman et al. [3] and Edmonson [4], where BCS is determined on a five-point scale with 0.25 increments, and 1 represents an extremely lean cow and 5 represents an extremely obese cow.

Over the years, several studies have shown the effect of different BCS conditions around calving on milk production, reproduction, metabolic profile, and the occurrence of diseases in dairy cows. In such studies, extremely obese cows had lower dry matter (DM) intakes after calving [2,5] and higher incidence of diseases, such as ketosis, fatty liver, milk fever, and displaced abomasum [4,6]. Very lean cows can have their production potential reduced in addition to being more susceptible to diseases [4].

Recently, research has shown that metabolic and environmental conditions during pregnancy, particularly the last two months of pregnancy, affect not only the subsequent lactation of the cow but also the performance and health of its offspring. The effect of some special maternal metabolic conditions, such as cows submitted to heat stress during the dry and pre-calving periods has been very well elucidated recently in studies such as those by Tao et al. [7], Monteiro et al. [8], and Laporta et al. [9]. However, there are still many doubts about how maternal metabolism, calving season, and parity can interact and alter the metabolism of offspring, thus altering the ability to express productive potential and genetic improvement [1]. Only a few studies have evaluated maternal body condition score effects in dairy cows [10,11,12] and long-term effects, such as first lactation performance, are not elucidated.

Therefore, the objective of this study was to evaluate the impact of BCS classes on calving, parity, and calving season effects on the performance of dairy cows and their offspring. We hypothesized that higher BCS at calving would negatively impact offspring and cow’s performance, and also those multiparous daughters performed better than nulliparous ones, and that the calving season will impact both mothers’ and daughters’ performance.

## 2. Materials and Methods

### 2.1. Database Selection

The dataset used in this study came from a commercial farm located in Southern Brazil (Latitude: 24°47′28″S, Longitude: 50°00′43″W). Data from 521 Holstein cows that calved a female calf and had their BCS evaluated at calving were used for this prospective cohort study. Dry and pre-partum cows were housed in a compost barn, and post-partum and lactating cows were housed in a free stall. Until weaning, calves are kept in individual shelters, and after this period, they are raised collectively. Typically, female calves are weaned around 90 days of age. These calves are fed with pasteurized waste milk in a step-up, step-down system providing 6–8 L of milk per day, and they also received a typical commercial starter and water during the pre-weaning period.

For the database building and editing, the calving information of cows whose offspring were female and had their BCS evaluated on the calving day was collected. The BCS was accessed immediately after calving, during colostrum milking by a trained farm employer. These cows were then separated into five BCS classes: class 1: <3.0 (*n* = 19), class 2: 3.0–3.25 (*n* = 134), class 3: 3.5–3.75 (*n* = 160), class 4: 4.0–4.25 (*n* = 142) and class 5: >4.25 (*n* = 66). The BCS assessment was carried out on the farm by a trained employee. The milk yield of the dams was collected at 100 (*n* = 408) and 305 (*n* = 403) days of lactation. Calvings from twins and stillbirths were not included in the analysis.

Regarding the female calves, the following variables were evaluated: birth weight, weaning weight, average daily gain (ADG) until weaning, the number of artificial inseminations (AI) per pregnancy of the daughters and age at first calving as well as the milk yield of the daughters at 100 and 305 days of lactation. The variable “culling rate until weaning” was defined by adding the animals that died or left the herd until the weaning date. The “total culling rate” variable includes calves and heifers that died or left the herd from birth until the first calving.

The calving season was considered using calvings between the first and the last day of each season; fall: from 21 March to 21 June, average temperature 16 °C; winter: from 21 June to 23 September, average temperature 14 °C; spring: 23 September to 21 December, average temperature 18.5 °C; summer: From 21 December to 21 March, average temperature 23 °C.

### 2.2. Statistical Analysis

The study was designed as a prospective cohort study, and BCS class, calving season, and parity were included as fixed effects in the model. The variables with normal distribution, tested by Shapiro Wilk test, such as birth weight, weaning weight, ADG until weaning and age at first calving as well as the mothers’ and daughters’ 100-day and 305-day milk yields, were analyzed using the MIXED procedure of SAS (v. 9.4). The variables not normally distributed (number of AI per pregnancy of the daughters, culling rate until weaning, and total culling rate) were analyzed using the GLIMMIX procedure of SAS (v. 9.4), including the fixed effects of BCS class, calving season, and parity.

For results interpretation and discussion, effects were declared significant when *p* ≤ 0.05. Tendencies were declared when *p* > 0.05 and *p* ≤ 0.15.

## 3. Results

Daughters from primiparous cows were born lighter, weighing an average 2.3 kg less (*p* < 0.01, Table 1) than multiparous daughters. However, there was no difference (*p* > 0.10) in weaning weight, due to the trend (*p* = 0.06) of greater average daily gains for the primiparous cows’ daughters. There was no difference in age at first calving for primiparous and multiparous daughters, nor in their reproductive performance (*p* = 0.84 and *p* = 0.20, respectively). Unsurprisingly, primiparous cows had lower milk yields (*p* < 0.01) than multiparous ones; they produced 933 kg and 2128 kg less milk yield in 100 and 305-day periods than multiparous cows did. There was no effect (*p* > 0.10) of parity on the culling rate until weaning; however, for total culling, a tendency (*p* = 0.11, Table 1) towards a greater culling rate of daughters from multiparous dams was detected.

Calves born in winter had better performance than those born in other seasons, a fact demonstrated by the higher birth weight (*p* < 0.01) and greater weight at weaning (*p* < 0.01) compared to the other seasons (Table 2). Calves born in the winter also showed higher ADG value until weaning (*p* = 0.03) than calves born in summer. Calves born in the winter also had a lower age at first calving (*p* < 0.05) than calves born in spring and summer, but the number of AI per pregnancy was not affected by calving season (*p* = 0.20).

Cows that calved in winter produced more milk in 305 days of lactation (*p* < 0.01) than cows that calved in the remaining seasons. Likewise, the daughters of the cows that calved in spring and winter also produced more milk in their first lactation (*p* < 0.01). There was a tendency for a lower culling rate until weaning for calves that were born in the winter and spring seasons (*p* = 0.11, Table 2).

The BCS class had an impact (*p* < 0.01) on calf birth weights (Table 3 and Figure 1); daughters from class 1 cows (BCS < 3.0) were lighter (38.01 ± 1.0 kg) than the calves from class 5 cows with BCS > 4.25 (41.9 ± 0.57 kg). The daughters born from cows with intermediate BCS values (BCS from 3.0 to 4.25, classes 2, 3, and 4) had intermediate birth weights. However, this difference disappeared at weaning (*p* = 0.20, Table 3).

Cows in class 1 had the shorter gestation length, (270 days, *p* = 0.02, Table 1), while classes 2 and 3 had intermediate values (274.6 and 274.5 days, respectively). The longer gestation length was for the obese cows, with BCS > 4.25 (277.4 days).

There was a tendency (*p* < 0.06, Table 3) for lower milk yields at 305 days of lactation in class 1 cows (BCS < 3.0) versus class 4 and 5 cows (BCS ≥ 4.0). Daughters’ milk yields were not affected (*p* = 0.99) by maternal BCS class.

We found a tendency for the effects of BCS on the culling rate until weaning (*p* = 0.12, Table 3). Calves from dams with BCS < 3 (Class 1) had three times higher (31.8%) culling rate until weaning compared to calves from cows with BCS values of 3.0–3.25 (class 2), which had a culling rate of 9.6% (Figure 2). Maternal BCS had no influence (*p* > 0.10) on both age at first calving and reproductive performance.

## 4. Discussion

Comparing the offspring of multiparous and primiparous cows, multiparous daughters were born smaller, but reached the size of the other ones at weaning. The smaller body structures of heifers and the fact that they are still growing [13] explain the lower weights of the offspring from first lactation cows. In addition, the use of calving-ease bull semen in nulliparous cows is a common practice in many dairy farms.

As expected, primiparous cows produced less milk than multiparous. The incompletely developed mammary glands, combined with the demand for nutrients to complete the primiparous body’s development, are the main factors responsible for the lower production in the first lactation. According to Stott [13], a cow’s maximum yield occurs in the third or fourth lactation. However, the average productive life of a modern cow today in the USA is around 2.5–3.0 years after the first calving [14], demonstrating the importance of optimizing the body size of heifers so they can achieve greater yields in the first lactation. Poczynek et al. [15] suggested that the body weight at calving is more impacting on milk performance than the age at calving and that primiparous cows heavier at first calving produced more milk without negative impacts on reproduction performance.

From the point of view of production and quality of maternal colostrum, the tendency towards greater culling of daughters from multiparous cows is a contradiction, since mature cows are known to produce a greater volume of colostrum, a fact that may reflect in better passive immunity transfer for the calf, resulting in less risk of culling. However, Gonzalez-Recio et al. [11] found that heifers born of dams that were lactating while pregnant produced 53 kg less milk, left the herd earlier, and were metabolically less efficient. This is possibly due to the competition for nutrients with milk production that calves born from multiparous cows experience during pregnancy, which calves born from nulliparous cows do not experience. Another possibility is the intentionally greater retention of calves from primiparous cows because they typically represent greater genetic progress than calves from older cows. The offspring reproductive performance was not changed by the maternal parity, since age at calving and breeding per pregnancy number was not different.

Calves born in winter had better performance than those born in other seasons. Several studies have shown that daughters of cows subjected to heat stress during the dry and pre-parturition period have lower weights at birth and remain lighter at weaning [7,8,9,16]. Heat stress at the end of pregnancy can disrupt uterine circulation, decreasing the flow of nutrients to the calf, in addition to a predisposition to release inflammatory cytokines and increase the level of maternal and, consequently, fetal cortisol. These facts together lead to less intestinal colostrum absorption capacities and worse inflammatory responses in calves that suffer heat stress in the uterus [9]. We also found that calves born in winter had lower age at calving, a fact that can be explained by the better performance in early life showed by the high average daily gain during the pre-weaning period, since it was not difference in the breeding per pregnancy number

Cows that calved in winter and its daughters produced more milk in 305 days of lactation. Tao et al. [7] and Monteiro et al. [8] compared the performance of cows that spent the dry and pre-delivery period in the summer of the American state of Florida without any tool for mitigating heat stress with a group of contemporary cows that had fans and sprinklers in the feedbunk line. Cooled cows produced an average of 6.3 kg/d more milk compared to those without any tools to relieve heat stress. Regarding calves born from these cows, there was an increase of 5 kg/d in the first lactation of calves whose mothers had their thermal stress relieved. The authors continued this assessment and realized that, even in the third generation, the granddaughters of the cooled cows produced 3 kg/d more compared to those whose grandmothers were subjected to thermal stress. The above facts can explain the effects of calving season in both cows and their calves as found in the present study.

In the present study, we found a tendency for greater milk production for obese cows. The lowest production was found for those with a low BCS. Over the years, several studies have shown the effect of different BCSs around calving, on milk production, reproduction, metabolic profile, and disease occurrence in dairy cows. The pre-calving BCS is negatively correlated with DM intake after calving [9,13]. Therefore, over-conditioned cows tend to have greater mobilization of their reserves in the postpartum period, a fact that may predispose the liver to overload to metabolize these reserves. This leads to an excessive increase of non-esterified fatty acids in circulation, high production of ketone bodies, and the occurrence of the most undesired pathway: accumulation of fat in the liver parenchyma [17]. Thus, some studies show that overweight cows may be more predisposed to postpartum diseases [3,17]. However, a review by Roche et al. [17] shows that very lean cows are also at greater risk of succumbing to diseases during the transition period. Therefore, very lean cows are not only more susceptible to diseases but are also less productive. However, although obese cows are more likely to get sick, they are more productive.

Daughters of lean cows (BCS < 3) had lower birth weights than the other classes, while daughters of obese cows (BCS > 4.25) were heavier. The daughters born from cows with an intermediate BCS (Class 3) had intermediate weights. These data contrast with those presented by Lopes et al. [12] who, when comparing the weights of daughters of cows with normal body condition (BCS 3.25) with obese cows (BCS > 3.75), report that the daughters of obese cows were on average 2 kg lighter at birth, but these authors did not show the dam’s gestation length. The lower weights of offspring of lean cows could be due to a lower feed intake in this category. In this database, we do not have information about the occurrence of concomitant diseases such as, for example, hoof injuries, which are known to lead an animal having fewer visits to the feeding trough and, consequently, a reduction in feed intake. We also showed that lean cows had a 7-day shorter gestation period than obese cows. These extra days in utero could lead to higher birth weights for calves born to obese cows, as gestation length followed the same pattern as birth weight in the dataset.

On the other hand, the higher birth weights of calves born from obese cows may be related to the energy metabolism of these cows. Weber et al. (2016) report that, although the state of insulin resistance is a homeorhetic mechanism that most high-producing cows face, cows with high energy reserves have more pronounced insulin resistance. Thus, as already documented in human medicine, children of women who have gestational diabetes have greater birth weights [18,19]. Future studies with data on metabolites such as insulin and glucose in periparturient cows and the relationship with offspring metabolism may help to better elucidate this issue. Interestingly, although the daughters of obese cows were heavier at birth, they weaned numerically lighter, losing their potential advantage, while daughters of lean cows reached the average weight of the group at weaning.

Calves born of cows with a normal BCS had a lower culling rate until weaning compared with offspring from over-conditioned or lean cows. Ling et al. [20] evaluated the influence of metabolic stress at the end of pregnancy on the inflammatory response of calves born from the cows evaluated. Calves exposed to high maternal levels of non-esterified fatty acids (NEFA) had greater oxidative stress and higher concentrations of inflammatory markers during lactation. The authors concluded that metabolic stress in dairy cows during late pregnancy was associated with differences in the offspring’s immune and metabolic systems during the first month of life. This finding suggests that health challenges, such as diarrhea and pneumonia (the major diseases found on most farms), can be affected by the maternal metabolic status [1]. In human medicine, high maternal inflammation during gestation was positively associated with respiratory tract infections in children during the first 14 months of life [21]. Lopes et al. [12] found that daughters of obese cows had worse response fronts to an ex vivo challenge using *E. coli* LPS than normal cows, but the authors did not evaluate daughters from cows with low body condition scores. Although mortality was higher for heifers born to lean cows, we found no difference in the productive or reproductive performance of the heifers that remained in the herd. However, the greater culling until weaning decreases the number of females for replacement and can limit the growth of the farm.

The group of calves with the highest culling rate was the one from lean cows, those calves stayed one week less in the utero. Maternal undernutrition has been studied extensively in ruminants. Maternal nutrient supply below requirements is also associated with altered immune responses and increased morbidity and mortality risks, not only during the neonatal stage but also during the productive life of cows [22].

We encourage readers to consider the limitations of our study before generalizing our findings. One important limitation is that we did not have access to disease records during the cows’ pregnancy. Therefore, we could not associate the diseases with the offspring performance. We considered the BCS mensuration at calving day to organize the classes according to maternal energy reserves. However, the farm did not measure the BCS at dry-off. This limited us in accessing the body condition score change during the dry period and postpartum and offspring performance.

## 5. Conclusions

The results suggest that maternal and environmental factors, such as calving season and parity, in addition to the metabolic condition, herein reported as BCS classes, are associated with different offspring performances. The nutritional and management practices for lactating cows can cause effects in the subsequent lactation and long-term effects in the calves. The genetic improvement of the herds can be impaired or masked by adverse conditions in the intra and extra-uterine calf’s life. However, the physiological mechanisms of the uterine environment that influences performance remain not fully elucidated.

## Figures and Tables

**Figure 1 animals-13-00596-f001:**
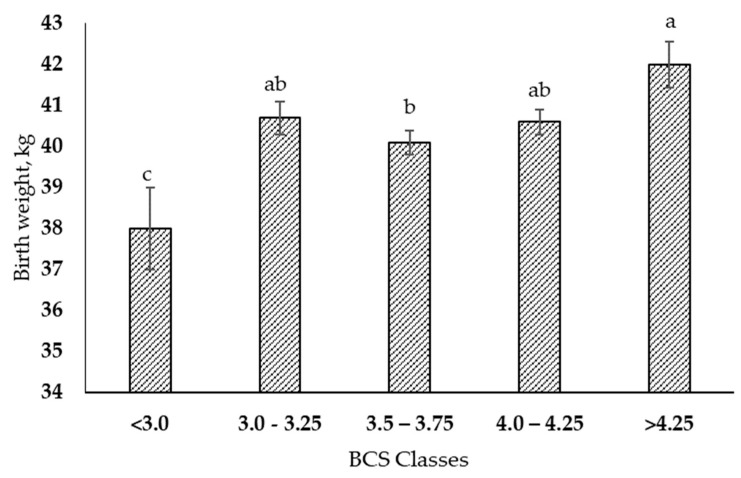
Effect of maternal body condition score on the birth weights of their offspring. ^a–c^ Bars with different superscripts differ significantly (*p* < 0.05). Error bars indicate the standard error of the mean.

**Figure 2 animals-13-00596-f002:**
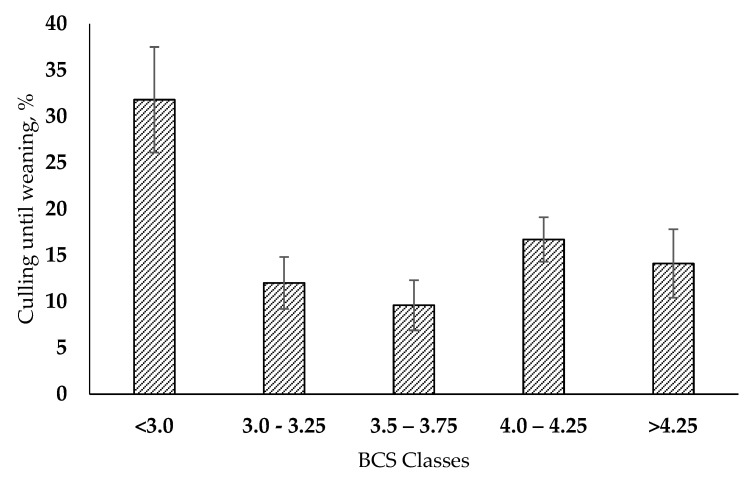
Effect of maternal body condition score on the culling rates until weaning of their offspring. Error bars indicate the standard error of the mean.

**Table 1 animals-13-00596-t001:** Effect of parity in the performance of cows and their offspring.

Item	Parity ^5^		*p*-Value
	Primiparous	Multiparous	
Birth weight, kg	39.1 ± 0.42	41.4 ± 0.29	<0.01
Weaning weight, kg	121.5 ± 1.67	120.4 ± 1.58	0.48
Average daily gain, g/d	910.0 ± 0.01	880.0 ± 0.01	0.06
Age at first calving, days	718.4 ± 13.9	721.1 ± 8.9	0.84
Dam 100-d MY, kg ^1^	3495.0 ± 139	4428.0 ± 92	<0.01
Dam 305-d MY, kg ^2^	8633.0 ± 363	10,761.0 ± 249	<0.01
Daughter 100-d MY, kg ^3^	3310.0 ± 182	3377.0 ± 113	0.84
Daughter 305-d MY, kg ^4^	8480.0 ± 815	8371.0 ± 489	0.89
Total culling, %	17.8 ± 2.4	24.5 ± 1.5	0.11
Culling until weaning, %	15.2 ± 1.8	16.3 ± 2.7	0.79
Breedings per pregnancy, n	1.8 ± 0.1	2.1 ± 0.7	0.20

^1^ Milk yield during the first 100 days of lactation of mothers; ^2^ Milk yield during the first 305 days of lactation of mothers; ^3^ Milk yield during the first 100 days of lactation of daughters; ^4^ Milk yield during the first 305 days of lactation of daughters. ^5^ Primiparous = 169, multiparous = 352.

**Table 2 animals-13-00596-t002:** Effect of the calving season on the performance of cows and their offspring.

Item	Calving Season ^5^				*p*-Value
	Winter	Spring	Summer	Fall	
Birth weight, kg	41.3 ^a^ ± 0.5	39.1 ^b^ ± 0.5	40.1 ^b^ ± 0.98	39.9 ^b^ ± 0.4	<0.01
Weaning weight, kg	125.8 ^a^ ± 2.07	120.1 ^b^ ± 1.8	117.9 ^b^ ± 1.50	121.5 ^b^ ± 1.9	<0.01
Average daily gain, g/d	930.0 ^a^ ± 0.02	900.0 ^ab^ ± 0.01	870.0 ^b^ ± 0.07	880.0 ^ab^ ± 0.02	0.03
Age at first calving, days	694.0 ^b^ ± 12	737.0 ^a^ ± 15	736.0 ^a^ ± 115	710.0 ^ab^ ± 11	0.04
Dam 100-d MY, kg ^1^	3873.0 ± 177	4050.0 ± 153	3933.0 ± 115	3929.0 ± 138	0.82
Dam 305-d MY, kg ^2^	10,860.0 ^a^ ± 467	9547.0 ^bc^ ± 398	96,119.0 ^b^ ± 299	8763.0 ^c^ ± 363	0.01
Daughter 100-d MY, kg ^3^	3377.0 ± 187	3413 ± 189	3321.0 ± 191	3547.0 ± 137	0.68
Daughter 305-d MY, kg ^4^	9214.0 ^ab^ ± 776	9960.0 ^a^ ± 860	7761.0 ^bc^ ± 847	6766.0 ^c^ ± 639	0.03
Total culling, %	24.0 ± 2.9	19.0 ± 2.8	19.0 ± 2.1	22.0 ± 2.2	0.80
Culling until weaning, %	12.2 ± 3.19	12.7 ± 3.2	16.0 ± 3.6	23.9 ± 2.4	0.11
Breedings per pregnancy, n	1.8 ± 0.1	2.1 ± 0.1	2.2 ± 0.1	1.6 ± 0.1	0.20

^1^ Milk yield during the first 100 days of lactation of mothers; ^2^ Milk yield during the first 305 days of lactation of mothers; ^3^ Milk yield during the first 100 days lactation of daughters; ^4^ Milk yield during the first 305 days of lactation of daughters. ^5^ Winter n = 82, spring n = 98, summer = 188, fall = 153. ^a–c^ Means in the same row with different superscripts differ significantly (*p* < 0.05).

**Table 3 animals-13-00596-t003:** Effect of body condition score at calving on the performance of cows and their offspring.

Item	BCS Classes ^5^					*p*-Value
	<3.0	3.0–3.25	3.5–3.75	4.0–4.25	>4.25	
Birth weight, kg	38.01 ^c^ ± 1.0	40.7 ^ab^ ± 0.4	40.1 ^b^ ± 0.3	40.6 ^ab^ ± 0.3	41.9 ^a^ ± 0.57	<0.01
Weaning weight, kg	121.7 ± 5.04	121.2 ± 1.54	121.6 ± 1.38	122.2 ± 1.48	118.0 ± 1.98	0.20
Average daily gain, kg/d	941.0 ± 0.05	894.0 ± 0.01	903.0 ± 0.01	901.0 ± 0.01	850.0 ± 0.02	0.20
Age at first calving, days	756.0 ± 29	721.0 ± 10	713.0 ± 10	711.0 ± 17	694.0 ± 19	0.46
Dam 100-d MY, kg ^1^	4019.0 ± 337	4101.0 ± 123	3847.0 ± 111	3965.0 ± 131	3873.0 ± 176	0.57
Dam 305-d MY, kg ^2^	8335.0 ± 933	9745.0 ± 315	9563.0 ± 285	10,463.0 ± 336	10,380.0 ± 450	0.06
Daughter 100-d MY, kg ^3^	3331.0 ± 383	3440.0 ± 122	3274.0 ± 123	3518.0 ± 249	3509.0 ± 264	0.77
Daughter 305-d MY, kg ^4^	8829.0 ± 1523	8288.0 ± 560	8307.0 ± 560	8324.0 ± 1072	8380.0 ± 1314	0.99
Dam gestation length	270.4 ^c^ ± 1.84	274.6 ^c^ ± 0.73	274.5 ^b^ ± 0.69	275.0 ^ab^ ± 0.84	277.0 ^a^ ± 1.10	0.02
Total culling, %	34.3 ± 5.4	18.4 ± 2.3	20.3 ± 2.0	21.5 ± 2.2	20.0 ± 3.2	0.96
Culling until weaning, %	31.8 ± 5.7	12.0 ± 2.8	9.6 ± 2.7	16.7 ± 2.4	14.1 ± 3.7	0.12
Breedings per pregnancy, n	2.15 ± 0.2	1.95 ± 0.2	1.67 ± 0.1	2.12 ± 0.1	1.95 ± 0.1	0.44

^1^ Milk yield during the first 100 days of lactation of mothers; ^2^ Milk yield during the first 305 days of lactation of mothers; ^3^ Milk yield during the first 100 days of lactation of daughters; ^4^ Milk yield during the first 305 days of lactation of daughters. ^5^ Class 1: < 3.0 n = 19, class 2: 3.0–3.25 n = 134, class 3: 3.5–3.75 n = 160, class 4: 4.0–4.25 n = 142 and class 5: >4.25 n = 66. ^a–c^ Means in the same row with different superscripts differ significantly (*p* < 0.05).

## Data Availability

Data will be made available upon reasonable request to the corresponding author.

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
