# Peer review of "Associations of Body Condition Score at Calving, Parity, and Calving Season on the Performance of Dairy Cows and Their Offspring"

_animals, 2023, doi:10.3390/ani13040596_

Round 1

Reviewer 1 Report

This study has been performed well, is clearly written and provides some useful information. The following points do, however, need addressing.

L91-93. My main concern is over the BCS scoring system. From the data provided, hardly any of the cows were of BCS<3 at calving and many were greater than 4.25. These figures do not tally well with the normal expectations in Holstein cows. Of the papers cited, Wildman et al. give mean BCS at drying off and early lactation of 3.37 and 2.52 on a 5 point scale. Optimum BCS at calving is generally considered to be in the range 3-3.5. The values in this paper therefore seem to be on the high side. Was the scoring done before or after calving?

Provide more information on how the seasons were defined and their differences in temperature.

L114. I think there is a typing error and it should be P>0.05.

Include the n per group in Tables 1-3.

L155. Should this be BCS4?

Fig 1. Add P values.

L172. The authors mention use of different semen on the primiparous cows. Would it be possible to provide information on the respective genetic merit of sires used for PP and MP dams?

Reviewer 2 Report

It was a very nice and interesting article to read; please consider the following edits. Thank you

Line 47-48: What type of changes in tissue structure and function? Please add details in the text.

Line 52-52: “allows the estimation of adipose tissue reserves in cows.” Please cite this statement.

Line 62: Please delete the word ‘some’.

Line 62-64: Please cite this sentence. And please check throughout the manuscript and cite the statements taken from other researchers that you have not cited.

Line 81-83: Was this a retrospective or prospective study? Please describe at the start of the methodology and in the abstract as well. At the moment, it is unclear.

Line 90-91: Here, it seems that you have analyzed farm records, which means it was a retrospective study. But I am not sure till now. Please clearly write your study design at the start of the methods.

Line 105: “The study was designed as a prospective cohort study,” Now I know you meant early. Please add this statement in the abstract and at the start of the methods part.

Lines 126,142,162: Please make all the decimal places similar. Number of digits after the decimal point should be same. It looks good.

Line 192-193: Please start the discussion paragraph by telling your findings to the reader.

Lines 199-202 and 249-250: Please tell us your results at the start of each discussion paragraph and then explain it with the help of previous literature.

Line 268: Please add the limitations of your study as the last paragraph of your discussion.

Please discuss all of your results (Table 1, 2, and 3) in the discussion.
